# A New Perspective on Traffic Flow Prediction: A Graph Spatial-Temporal Network with Complex Network Information

Zhiqiu Hu [1,2] , Fengjing Shao [2] and Rencheng Sun [2,*]

1 School of Automation, Qingdao University, Qingdao 266071, China
2 College of Computer Science and Technology, Qingdao University, Qingdao 266071, China
* Correspondence: src@qdu.edu.cn; Tel.: +86-13184130501

**Abstract:** Traffic flow prediction provides support for travel management, vehicle scheduling, and intelligent transportation system construction. In this work, a graph space–time network (GSTNCNI), incorporating complex network feature information, is proposed to predict future highway traffic flow time series. Firstly, a traffic complex network model using traffic big data is established, the topological features of traffic road networks are then analyzed using complex network theory, and finally, the topological features are combined with graph neural networks to explore the roles played by the topological features of 97 traffic network nodes. Consequently, six complex network properties are discussed, namely, degree centrality, clustering coefficient, closeness centrality, betweenness centrality, point intensity, and shortest average path length. This study improves the graph convolutional neural network based on the above six complex network properties and proposes a graph spatial–temporal network consisting of a combination of several complex network properties. By comparison with existing baselines containing graph convolutional neural networks, it is verified that GSTNCNI possesses high traffic flow prediction accuracy and robustness. In addition, ablation experiments are conducted for six different complex network features to verify the effect of different complex network features on the model's prediction accuracy. Experimental analysis indicates that the model with combined multiple complex network features has a higher prediction accuracy, and its performance is improved by 31.46% on average, compared with the model containing only one complex network feature.

**Keywords:** traffic flow prediction; GSTNCNI; complex network feature information

## 1. Introduction

As a physical fundamental of any country, the road transportation network plays a vital role in both social and economic development. However, with recent rapid socioeconomic development, car ownership continues to rise, which in turn places tremendous pressure on road traffic, and traffic congestion is becoming increasingly problematic [1]. Optimizing and transforming the traffic network and intelligent management are effective ways to alleviate traffic congestion. Road traffic network analysis and research are of great importance and strongly support intelligent transportation.

Complex networks which describe the relationship between different individuals have become an emerging research hotspot in recent years. Individuals are usually considered to be network nodes and connecting edges if there is a specific relationship between them. Therefore, a large number of nodes and the connected edges between them constitute a complex network. Complex network theory has good applicability in several complex problem fields, including protein, citation, computer, and power networks. Transportation networks are also typical complex network structures [2–4]. Many studies have introduced complex network theory into transportation network analysis and research. Shen et al. [5] used complex network theory to analyze the metro-bus composite network in Chengdu, a small-world network with scale-free features, and analyzed its resistance to destruction

under random and deliberate attacks. Chang et al. [6] used complex network theory to analyze the Beijing rail transit network. Zhang et al. [7] established the main skeleton regional network and urban network of the national comprehensive three-dimensional transportation network by searching for important nodes with high centrality. They then analyzed their robustness based on proximity centrality, mediator, and PageRank centrality, respectively. Hu et al. [8] applied complex network theory to urban traffic congestion factor risk propagation. They proposed the concept of network node importance, based on complex network metrics, to classify network nodes and perform direct immune control on core nodes. It was found that this method can reduce the degree of traffic congestion factor risk and speed up risk recovery. Yang et al. [9] applied complex network theory to critical node identification and integrated the degree and clustering coefficients and neighbors to identify critical nodes and calculate the weights of degrees and clustering coefficients using entropy techniques. The experimental results from four real networks showed that the method can more effectively identify critical nodes. The above study only analyzes traffic complex network topology and features and does not discuss integration with intelligent transportation.

Intelligent transportation can effectively improve traffic congestion and operational efficiency through traffic control and inducement, while real-time and accurate traffic flow prediction is the key to achieving intelligent traffic management. Through various monitors distributed on the traffic network, we can obtain rich traffic data which researchers can apply to deep learning traffic flow prediction models [10,11].

Long short-term memory networks (LSTM), convolutional neural networks (CNN), and combined models have been successively applied to traffic prediction problems. Jia et al. [12] used deep belief networks (DBN) with LSTM to achieve traffic flow prediction under wet weather conditions in Beijing. Ma et al. [13] dynamically transformed traffic data into images describing spatial–temporal traffic flow relationships and used CNN to extract the spatial–temporal features in the image to realize the traffic speed prediction. LSTM is good at processing time-series data [14] and CNN is good at extracting Euclidean spatial data features [15]. Traffic data not only have a temporal correlation, but also have a spatial relationship between monitors, as the source of traffic data is an ir-regular graph structure belonging to non-Euclidean space. Therefore, to fully exploit the spatial correlation between data, a graph convolutional neural network (GCN) is introduced to traffic flow prediction. Zhao et al. [16] proposed a T-GCN model, which uses GRU to extract temporal correlations between data, and GCN to extract spatial correlations. Yu et al. [17] established an ST-GCN consisting purely of convolutional neural networks, which combines graph convolution and gated-temporal convolution. There are fewer parameters in this model, allowing it to be more effectively applied to large-scale datasets. On this basis, Guo et al. [18] proposed the MSTGCN model and applied it to highway traffic prediction. The MSTGCN consists of three independent components modeling the proximity dependence, daily cycle, and weekly cycle dependence of traffic data, with each component consisting of a standard 2-dimensional graph convolution. The ASTGCN model [19] is based on the MSTGCN but with an added spatial–temporal attention mechanism.

Complex network theory, as a new graph structure data analysis tool, can deeply analyze and mine the spatial relationships among monitoring stations. Tang et al. [20] used complex network theory to study traffic flow time series, providing a new perspective for traffic flow analysis. The complex network degree distribution can be fitted to a Gaussian function and the cumulative degree distribution can be fitted to an exponential function. The density and clustering coefficients reflect the changes in connectivity between nodes in complex networks, and the change results are consistent with the observed adjacency matrix graph results. Consequently, the range of critical thresholds can be determined, based on complex network theory. This work provides a new method for understanding the dynamics of traffic flow time series. Tang et al. [21] used the Lempel–Ziv algorithm to evaluate traffic flow data complexity over different time scales. To gain more insight

into the complexity and periodicity of the traffic flow time series, each day is considered a cycle, and each cycle is considered a single node, thus constructing a complex network from the traffic flow time series. The complex networks are analyzed according to various statistical properties, including average path length, clustering coefficient, average degree, and intermediate degree. The experimental results in the above paper show that complex networks are a practical tool for mining the dynamic variation characteristics of traffic flow time series. However, the paper only analyzes the traffic flow data through a complex network and does not apply it to traffic prediction.

In this research, complex networks and GCN are combined to predict traffic flow. Firstly, a traffic complex network model is established using highway traffic big data, the topological features of traffic road networks are then analyzed by complex network theory, and finally, the topological features are combined with graph neural networks to explore the role of topological features in complex networks in traffic flow prediction. The main motivations and contributions of this paper are as follows.

(1) In traffic flow forecasting, many forecasting models are highly accurate. However, a spatiotemporal prediction model, based on the graph structure, cannot consider multiple properties in the graph structure information, which would otherwise enable it to improve the prediction accuracy. This work mines and analyzes many features and information of the complex traffic network composed of traffic nodes and proposes a novel graph spatial–temporal model (GSTNCNI) that can merge the complex network feature information. The prediction times are 5 min, 30 min, and 60 min.

(2) The spatial convolution layer in GSTNCNI can fuse the complex network feature information to fully capture and mine the complex spatial correlation relationships between traffic nodes. The temporal convolutional layer in GSTNCNI is used to compute time-dependent time-series features. To overcome the limitations of traditional recurrent neural networks that do not support parallel computation and slow training speed, the temporal convolution layer uses a multilayer residual structure instead of the gating mechanism of recurrent neural networks.

(3) Comparison and ablation experiments are conducted on GSTNCNI using the PeMS dataset. The experimental results show that GSTNCNI is optimal, and exhibits a strong, stable traffic flow prediction performance.

## 2. Traffic Complexity Network

### 2.1. Traffic Complexity Network Construction

Complex network theory is used to model the traffic complex network to intensely analyze the traffic network topological features. Complex network modeling mainly involves two methods, Space L and Space P [22]. The Space L method abstracts monitoring stations as nodes and connects edges between two nodes through roads if they are directly adjacent. The Space P method also abstracts monitoring stations as nodes but connects edges between two nodes if they are reachable through roads. The complex network established by the Space L method is closer to the real traffic road network and is more conducive to the extraction of monitoring station spatial features. Consequently, this paper employs the Space L method to model the traffic network.

The traffic complex network can be defined as $G = (V, E, W)$, where the set $V = \{v_1, v_2, \ldots, v_n\}$ is the set of nodes, $n$ is the total number of nodes, and a monitoring station is abstracted as a node. The set $E = \{e_1, e_2, \ldots, e_m\}$ is the set of edges and m is the total number of edges when two monitors are directly adjacent, which corresponds to two nodes with edges between them, i.e., $e_k = (v_i, v_j) \in E$ and $v_i, v_j \in V$. Additionally, $W : E \rightarrow R$, for each edge $e_k$ of $G$, $W(e_k) = d(v_i, v_j)$ is the weight on the edge. $d(v_i, v_j)$ is the shortest distance between two points $v_i, v_j$. The shortest distance in the unweighted graph refers to the number of edges of the shortest path between two points, and the sum of the weights of the edges in the shortest path in the weighted graph is the shortest

distance between two points. The expression of the complex network adjacency matrix $A = \left\{a_{ij}\right\}_{n \times n}$. $a_{ij}$ is shown in Equation (1).

$$a_{ij} = \begin{cases} 1, \left(v_i, v_j\right) \in E \\ 0, \left(v_i, v_j\right) \notin E \end{cases} \tag{1}$$

The traffic California highway network is analyzed in this paper, with data obtained from the PeMS highway dataset [23]. The data are gathered from sensors distributed along the highway, and this study contains data downloaded from 157 sensors, including the sensor location coordinates as well as traffic flow, speed, and lane occupancy. The traffic flow is sampled at 5 min intervals, with each sensor obtaining up to 288 pieces of data a day. The coordinates data are used to model the highway network, while the flow data are used for traffic forecasting.

The nodes are numbered by removing duplicate locations and proximity at 157 monitoring points, which are then sorted according to latitude and then longitude. Subsequently, a neighbor table is built between the nodes according to the neighboring relationship, from which the road network model is finally formed. The final traffic complex network contains 97 nodes and 178 edges, and is shown in Figure 1. As seen in Figure 1, each node in the graph represents a physical sensor. Due to the large distance between physical sensors, the traffic network topology cannot be observed intuitively. Therefore, this paper represents physical sensors in the graph in the form of nodes that are connected in the topology graph according to the road connectivity between the sensors.

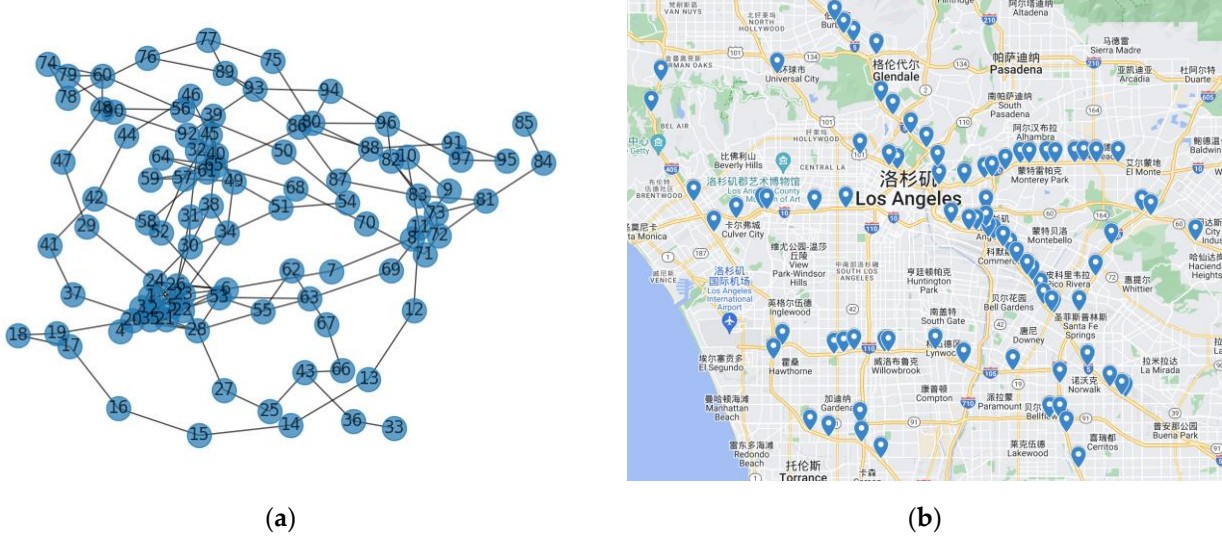

(**a**)                                        (**b**)

**Figure 1.** Traffic complex network structure and node location. (**a**) Traffic complex network structure; (**b**) The geographical location of the 97 monitoring stations.

### 2.2. Analysis of Complex Network Topological Features

The topological features of a complex network are parameters used to describe its structural features, and this paper analyzes the importance and the degree of mutual influence of nodes in the network. Statistics such as degree centrality, clustering coefficient [24], closeness centrality, betweenness centrality [25], point strength [26], and ASPLN [27] are introduced to characterize the node structural features. Among them, point strength and average shortest path length are the statistical parameters in the weighted graph.

#### 2.2.1. Degree and Degree Centrality

The degree of a node in a network indicates the number of times the point acts as an edge endpoint, and for simple graphs, the degree is the number of neighboring nodes. The greater the node degree, the more connected it is to other nodes and the greater its

importance. Degree centrality is obtained by normalizing the degree. Equations (2) and (3) are the formulas for degree and degree centrality, respectively.

$$K_i = \sum_{j=1}^{n} a_{ij} \tag{2}$$

$$DC_i = \frac{K_i}{n-1} \tag{3}$$

where $a_{ij}$ represents the connectivity between node $i$ and node $j$. If node $i$ is connected to node $j$, then $a_{ij} = 1$, otherwise $a_{ij} = 0$. $K_i$ represents the degree value of the $i$-th node. Degree centrality of the $i$-th node is $DC_i$. The node degrees of this highway network are shown in Figures 2 and 3, and the node color in Figure 2 represents the node degree value. The node with degree 1 is the road starting point, the node with degree 2 accounts for 45% at most, the degree values of the rest of the nodes are concentrated in 3, 4, and 5, and there are three nodes with degree values up to 12. The node degree of the network conforms to the power law distribution.

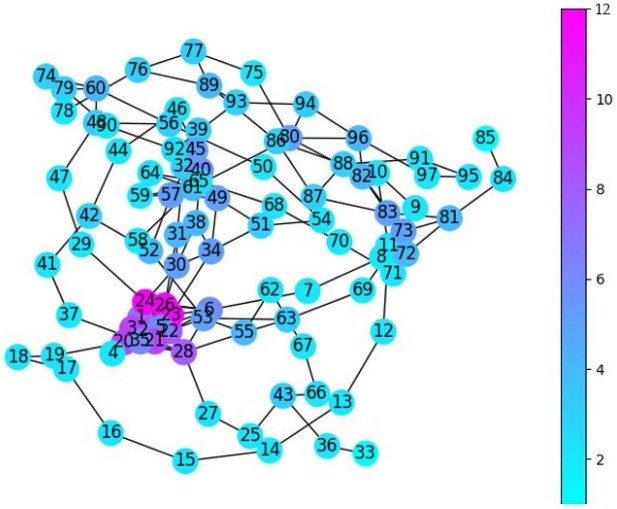

**Figure 2.** Distribution of node degree values of complex traffic networks.

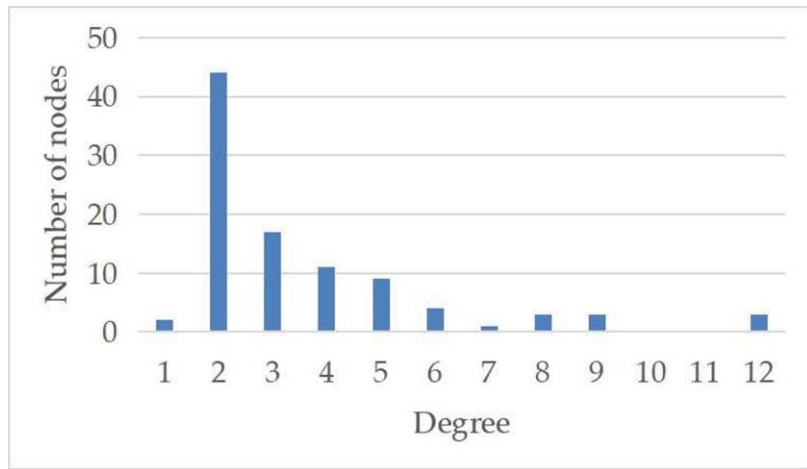

**Figure 3.** Node degree of traffic complex network.

### 2.2.2. Clustering Coefficients

The clustering coefficient represents the degree of node aggregation in a network and is characterized by the number of edges between neighboring nodes. Assuming that the

$i$-th node degree is $K_i$, there are at most $(K_i(K_i - 1))/2$ possible edges between $K_i$ nodes, and the clustering coefficient of a node is the ratio of the actual number of edges between $K_i$ nodes to the maximum number of possible edges, as given by Equation (4).

$$C_i = \frac{E_i}{(K_i(K_i - 1))/2} \tag{4}$$

where $E_i$ is the actual number of edges between $K_i$ nodes. Clustering coefficients of the $i$-th node are represented by $C_i$. Figure 4 displays the clustering coefficient of this high-speed road network. Compared with the urban road network, the highway network has fewer intersections and more 2-degree nodes. So, more than 50% of the nodes in the network have a clustering coefficient of 0, 25% have a clustering coefficient between 0.3 and 0.4, and only 10% of the nodes have a clustering coefficient over 0.5. The average road network clustering coefficient is 0.19, indicating that it is a low-aggregation network.

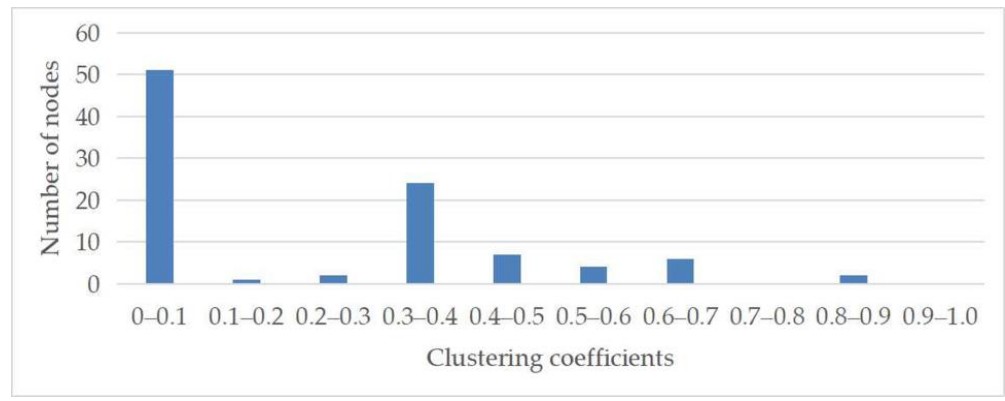

**Figure 4.** Clustering coefficients of traffic complex network.

2.2.3. Closeness Centrality

The centrality degree is calculated based solely on the network local characteristics, while the closeness centrality can be obtained based on the network global topology. Its value is defined as the reciprocal of the sum of the shortest distance between the node and all other nodes. The larger its value, the closer it is to the network geometric center. Facilities with high pedestrian traffic generally need to be built at locations with large closeness centrality. The normalized closeness centrality equation is shown in Equation (5).

$$CC_i = \frac{n - 1}{\sum_{j=1}^{n} d(v_i, v_j)} \tag{5}$$

where $d(v_i, v_j)$ represents the shortest distance between node $v_i$ and node $v_j$. $CC_i$ represents the closeness centrality of the $i$-th node. The highway network closeness centrality is shown in Figure 5. The minimum closeness centrality value is 0.095 and the maximum is 0.229. In total, 90% of the node closeness centrality lies between 0.1 and 0.2.

2.2.4. Betweenness Centrality

Betweenness centrality is a global statistic when two disjoint nodes in a network need to pass through other nodes to connect. The higher the number of times a node acts as an "intermediary", the greater its betweenness centrality. Betweenness centrality is defined as the ratio of the number of shortest paths through the node to the number of all the shortest paths in the network. The normalized betweenness centrality equation is Equation (6), where $g_{st}$ is the number of shortest paths between nodes $s$ and $t$, and $g_{st}(i)$ is the number of shortest paths between $s$ and $t$ through $i$. $[(n-1)(n-2)]/2$ is the maximum value of

the shortest path possible through the $i$ node. $BC_i$ represents the betweenness centrality of the $i$-th node.

$$BC_i = \frac{1}{[(n-1)(n-2)]/2} \sum_{s \neq i \neq t} \frac{g_{st}(i)}{g_{st}}$$

(6)

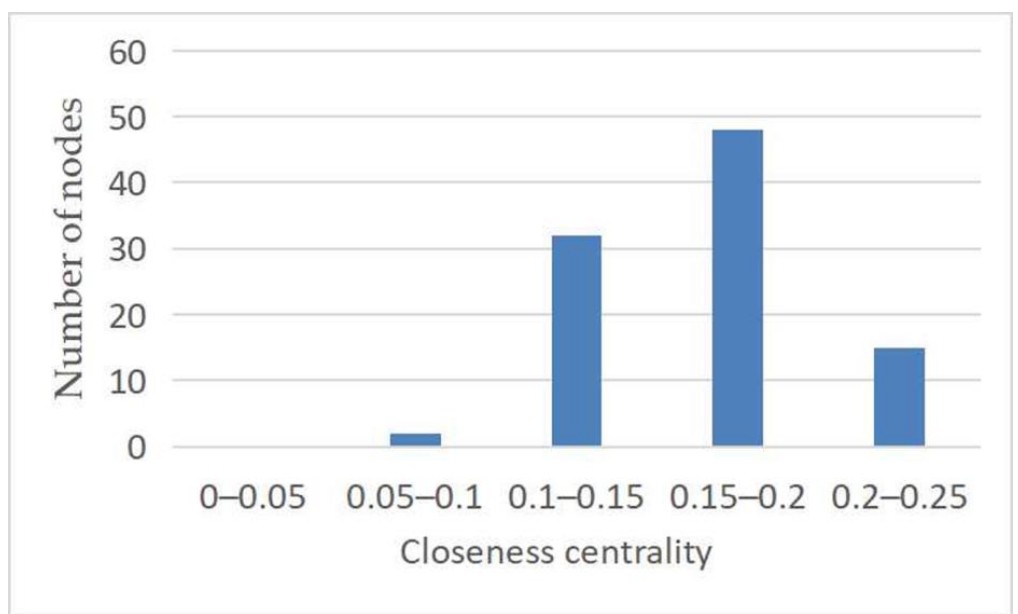

**Figure 5.** Closeness centrality of the traffic complex network.

The highway network betweenness centrality is shown in Figure 6. The betweenness centrality varies widely between nodes, from a minimum value of 0 to a maximum value of 0.26. In total, 82% of the nodes have intermediary centrality between 0 and 0.1.

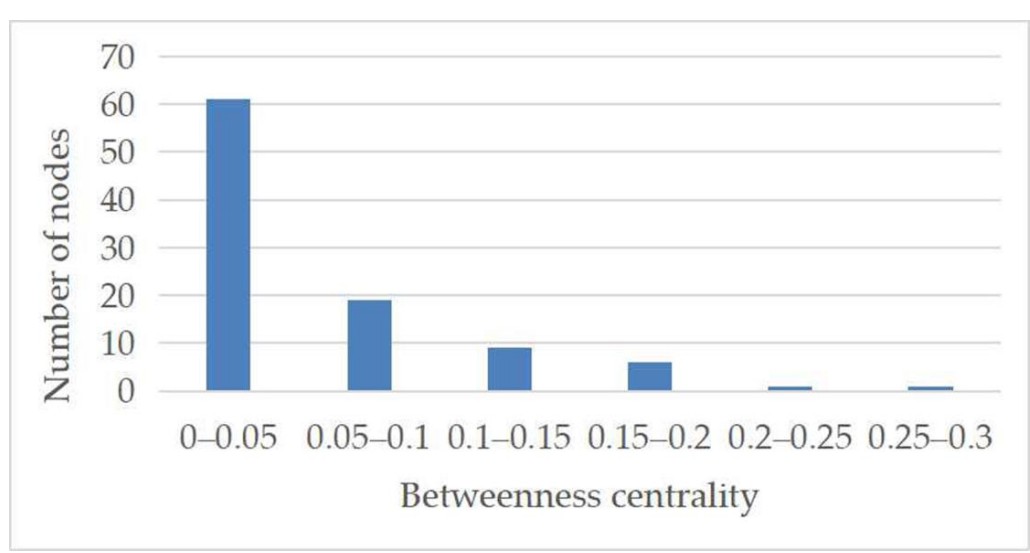

**Figure 6.** Betweenness centrality of the highway network.

### 2.2.5. Point Strength

The node degree reflects the node network structure characteristics while the point strength adds the node's associated edges' weight information to the degree value. Thus, the node characteristics reflected by the point strength are more accurate and comprehensive than the degree value. The point strength is defined as the sum of the weights of the



node edges and the adjacent nodes, and its formula is Equation (7). $S_i$ represents the node strength of node $i$. $w_{ij}$ represents the weight of the edge between node $i$ and node $j$.

$$S_i = \sum_{j \in \Gamma_i} w_{ij} \tag{7}$$

The highway network point strength is shown in Figure 7. The maximum value is 358 and the minimum value is 0.3. In total, 74% of the nodes have point strength between 0 and 50.

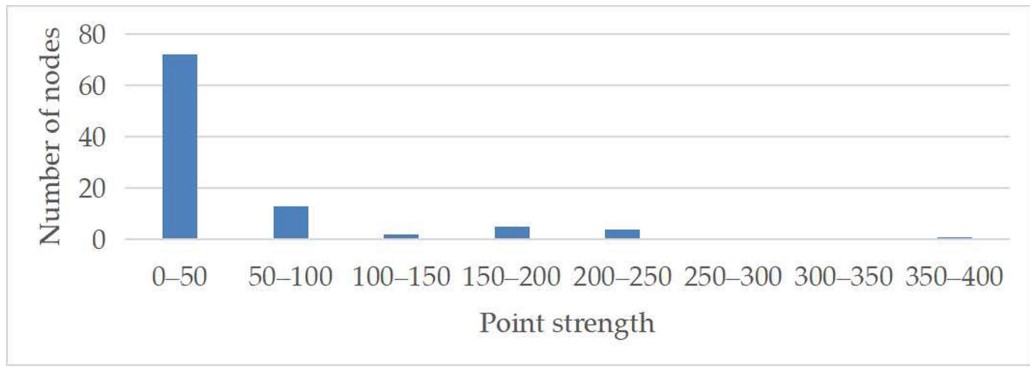

**Figure 7.** Point strength of highway network.

2.2.6. Average Shortest Path Length of Nodes

The average shortest path length of nodes (ASPLN) is defined as the average length of the shortest paths from that node to other nodes by adding the edge weight information in the network based on the closeness centrality. Its formula is Equation (8).

$$aspln_i = \frac{1}{n-1} \sum_{j=1, j \neq i}^{n} d(v_i, v_j) \tag{8}$$

Here, the shortest path length is the sum of the edge weights in the shortest path.

The highway network ASPLN is shown in Figure 8. The maximum value is 88 and the minimum value is 27. In total, 57% of the nodes have an ASPLN of between 20 and 40.

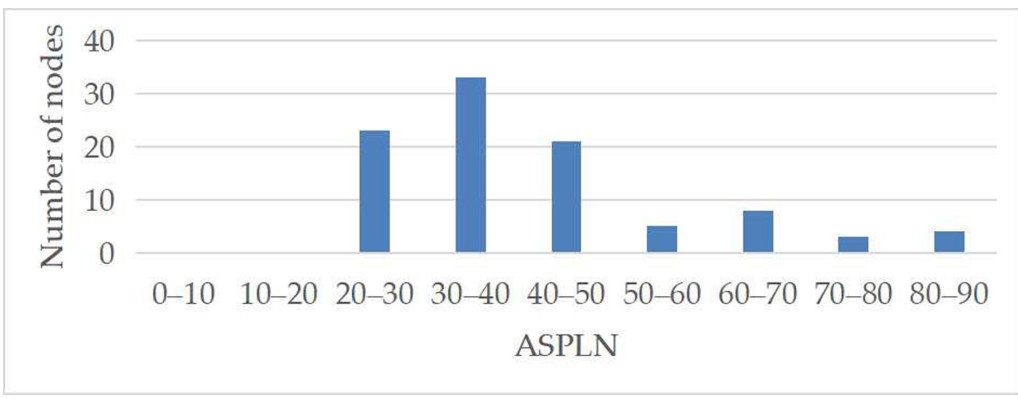

**Figure 8.** ASPLN of the highway network.

## 3. Deep Spatial–Temporal Model Design

Figure 9 shows the GSTNCNI model structure which consists of two main parts: the spatial convolutional layer for modeling the spatial correlation relationship between traffic nodes and the temporal convolutional layer for modeling the temporal data temporal dependence features. To highlight the impact of complex network features on the model

performance, we introduce the complex network feature calculation process in spatial relations. A detailed elaboration of the spatial and temporal convolutional layers is shown in Sections 3.1 and 3.2, respectively.

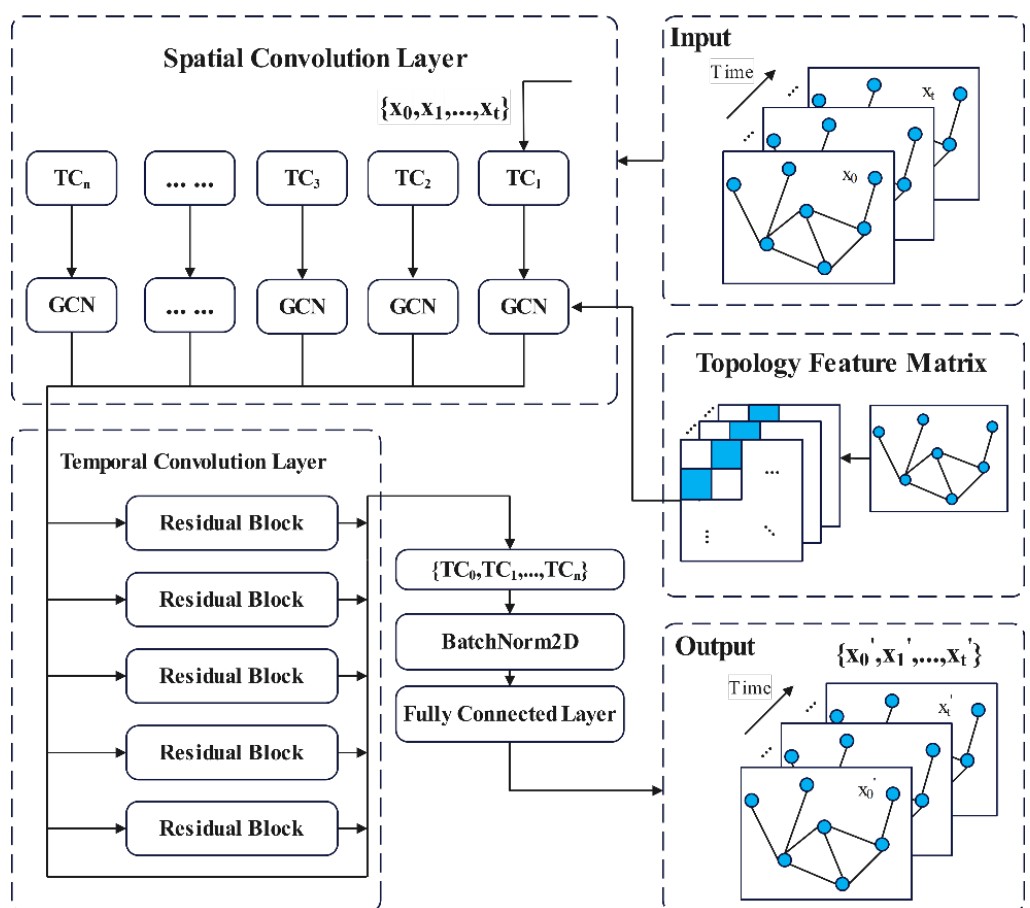

**Figure 9.** Structure of the GSTNCNI.

(1)    Input data periodicity

This paper divides the traffic flow data into 5 min time steps. As described in Section 2.1, the traffic network topology graph is defined as $G = (V, E, W)$, where $V = \{v_1, v_2, \ldots, v_n\}$ is the set of nodes in the traffic network topology graph. $|V| = n$ indicates that there are $n$ nodes in the topology graph. Let the input data time periods be $T$; $x_t^i \in \mathbb{R}$, $i \in [1, 2, \ldots, n]$, $t \in [1, 2, \ldots, T]$ represent the $i$-th node traffic flow size in the $t$-th time periods. $X_t \in \mathbb{R}^n$ represents the traffic flow of all nodes at the $t$-th time periods. $X = \{X_1, X_2, \ldots, X_T\} \in \mathbb{R}^{n \times T}$ represents all traffic flow data. Since the change of a node's traffic flow data is affected by the traffic flow change of the node in the previous period, this paper uses the historical traffic flow data of the previous hour, that is, 12 time periods, to predict the future (12 time periods) traffic flow. When the model predicts the traffic flow of all nodes 12 time periods after $t_p$, the input data of the model is defined as:

$$Input = \left\{ X_{t_p-11}, X_{t_p-10}, \ldots, X_{t_p} \right\} \in \mathbb{R}^{n \times 12} \tag{9}$$

(2)    Data Standardization

When training the traffic flow data, the large range of traffic flow values can lead to low traffic data prediction accuracy and slow gradient descent for the optimal solution. To resolve these issues, the Z-score normalization method is used in this paper and the data are processed to conform to standard normal distribution with the transformation function.

where $X$ is the real traffic data, $\mu$ is the mean of all the sample data, and $\sigma$ is the standard deviation of all the sample data. During the data standardization process, the traffic flow data mean and standard deviation need to be calculated. Standardization of the data can lessen the effect of some large variable values on the model.

$$X_{norm} = \frac{X - \mu}{\sigma} \tag{10}$$

The Dropout mechanism [28] is used to prevent the overfitting of the complex model in this paper. This mechanism simplifies the model by randomly discarding a part of the neuron nodes (feature detectors) which prevents the model from overfitting. Since the two neurons do not necessarily appear in a network, this can reduce the interaction between the neuron nodes. By randomly discarding part of the neuron nodes, the neurons can pay more attention to whether their feature detection is advantageous to the final result, rather than relying on the feature values detected by other neurons with a fixed relationship.

(3)    Output data periodicity

In addition, $Y_t^i \in \mathbb{R}$ indicates the $i$-th node traffic flow at a future moment $t$. The traffic flow of all nodes at the future moment $t$ is shown in Equation (3).

$$Output = \left( Y_{t_p+1}^1, Y_{t_p+2}^2, \dots, Y_{t_p+12}^n \right) \in \mathbb{R}^n \tag{11}$$

*3.1. Spatial Convolution Layer*

In the graph neural network process, it is essential that each node in the graph is influenced by its related neighbor nodes and changes its state until it reaches an equilibrium state. This influence becomes greater as the closeness of the relationship increases [29]. As mentioned above, the network topological features can describe the network-resultant features and reflect the role and importance of each node in the network. The graphical neural network introduces a modified version of the Laplacian matrix to incorporate the influence of neighboring nodes as well as its own nodes in the network into the calculation process. As shown in the Figure 10, the Laplacian matrix $L$ of a graph is defined as: $L = D - A$, where $D$ is the degree matrix. $A$ is the adjacency matrix [30]. The normalized Laplacian matrix is shown in Equation (12).

$$L_{sym} = D^{-\frac{1}{2}} L D^{-\frac{1}{2}} \tag{12}$$

| Graph | Degree matrix (D) | Adjacency matrix (A) | Laplacian matrix (D-A) |
|---|---|---|---|
|  | $\begin{bmatrix} 1 & 0 & 0 & 0 & 0 \\ 0 & 3 & 0 & 0 & 0 \\ 0 & 0 & 3 & 0 & 0 \\ 0 & 0 & 0 & 2 & 0 \\ 0 & 0 & 0 & 0 & 3 \end{bmatrix}$ | $\begin{bmatrix} 0 & 1 & 0 & 0 & 0 \\ 1 & 0 & 1 & 0 & 1 \\ 0 & 1 & 0 & 1 & 1 \\ 0 & 0 & 1 & 0 & 1 \\ 0 & 1 & 1 & 1 & 0 \end{bmatrix}$ | $\begin{bmatrix} 1 & -1 & 0 & 0 & 0 \\ -1 & 3 & -1 & 0 & -1 \\ 0 & -1 & 3 & -1 & -1 \\ 0 & 0 & -1 & 2 & -1 \\ 0 & -1 & -1 & -1 & 3 \end{bmatrix}$ |

**Figure 10.** The Laplacian matrix.

The problem of transferring node information is overcome in the Laplacian matrix, and the first-order Chebyshev approximation is used to simplify the calculation [31], as shown in Equation (13), where $\theta$ is the trainable parameter matrix, $D$ is the degree matrix, and $x$ is the input signal.

$$\theta *_g x = \theta \left( I_n + D^{-\frac{1}{2}} A D^{-\frac{1}{2}} \right) x \tag{13}$$

The matrix T is introduced as the network topological characteristic matrix. $t_{ii}$ can represent the six network topological features above, such as degree centrality, clustering

coefficients, closeness centrality of nodes, etc. Parameter initialization is an important part of parameter training and has a significant impact on the model accuracy.

$$T = \begin{pmatrix} t_{11} & \cdots & 0 \\ \vdots & \ddots & \vdots \\ 0 & \cdots & t_{nn} \end{pmatrix}; T_{sym} = \begin{pmatrix} \frac{t_{11}}{\sum_{i=1}^{n} t_{ii}} & \cdots & 0 \\ \vdots & \ddots & \vdots \\ 0 & \cdots & \frac{t_{nn}}{\sum_{i=1}^{n} t_{ii}} \end{pmatrix} \quad (14)$$

By normalizing the $T$ matrix to $T_{sym}$, the influence of different features is reduced in terms of magnitude, thus reflecting the relative importance of the node's features in the network. $\theta$ is initialized to the value of $T_{sym}$ matrix, thus realizing the topological properties of the network into the graph convolution process. $\theta$ already contains the relative-importance information of the node at the beginning, which can achieve faster convergence in the training process. This makes the results closer to the global optimal solution, as shown in Equation (14).

Different features correspond to different initialization parameter matrices, which in turn correspond to different GCN modules, and the modules corresponding to different features are fused by weighting and are trained centrally, as shown in Equation (15), where $n$ is the number of GCNs.

$$\theta *_g x = (\theta_1 \oplus \theta_2 \oplus \cdots \oplus \theta_n) *_g x \quad (15)$$

### 3.2. Temporal Convolution Layer

A special Temporal Convolution Layer (TCL) was designed in this paper to compute the data time-dependent features, as shown in Figure 11. The TCL is particularly useful in three ways. Firstly, it can retain all the historical information and compute long-term historical information using causal convolution. Secondly, it uses dilated convolution to extend the receptive field of the convolution process. The dilation rate varies according to a convex function, which allows the model to compute deep features without losing any local information due to an oversized receptive field. Finally, it is entirely composed of convolutional networks and uses a multilayer residual structure instead of the traditional "gate" structure. Furthermore, TCL overcomes the traditional RNN issues of not supporting parallel computation and slow training speed [32]. In general, TCL can effectively mine and model the data temporal features, unlike Convolutional Gated Recurrent Units (ConvGRU), which allows GRU to directly process image information by introducing convolutional operations.

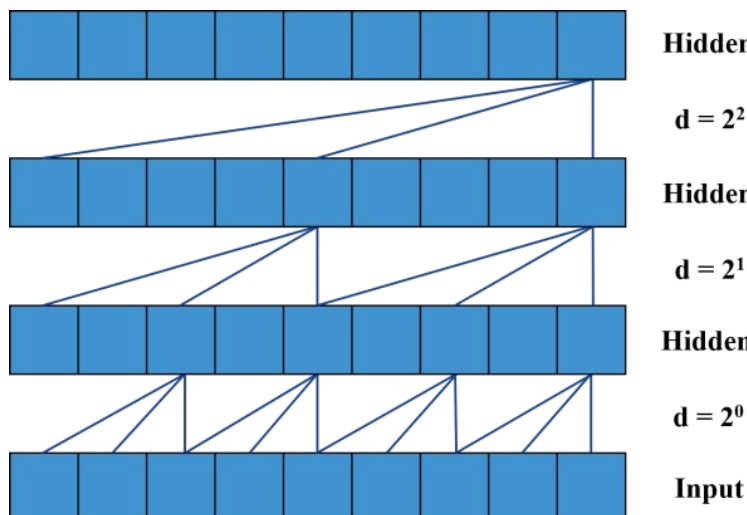

**Figure 11.** Structure of the Temporal Convolution Layer.

Causal convolution [33] in the TCL is used to obtain long-term historical information. The causal convolution formula is shown in Equation (16), where $X = (x_1, x_2, \ldots x_T)$ denotes the input sequence, $Y = (y_1, y_2, \ldots y_T)$ denotes the output sequence, and $F = (f_1, f_2, \ldots f_k)$ denotes the filter. The causal convolution focuses solely on historical information and ignores future information. In addition, the larger K is, the more historical information is obtained during the causal convolution.

$$y_t = \sum_{i=1}^{K} f_i \cdot x_{t-K+i} \tag{16}$$

A downsampling process is required by the model in deep neural networks to increase the receptive field and reduce the computational effort. However, downsampling also leads to a decrease in spatial resolution. We therefore consider the use of dilated convolution [34] to maintain a certain degree of spatial resolution while increasing the receptive field. Dilated convolution has a parameter that sets the dilation rate, meaning that the convolution kernel is filled with dilation rate. As a result, the dilation rate and size of the receptive field vary, enabling multiscale information to be acquired. The dilated convolution formula is shown in Equation (17), where d denotes the dilation rate, which changes as a convex function according to the depth of the network. Increasing either d or K can widen the receptive field range. However, the receptive field will be deepened with the network layers in a deep network so that there is no correlation between the information obtained from the long-distance convolution, resulting in a loss of local information. Consequently, we vary the dilation rate according to an exponential function of 2, as in the d variation process in Figure 11. This design pattern ensures that the perceptual field range in the deep network is limited to a certain extent, reducing the loss of local information.

$$y_t = \sum_{i=0}^{K-1} f_i \cdot x_{t-i\cdot d} \tag{17}$$

The residual structure is used instead of the traditional "gate" structure to diminish the training process complexity and increase the training speed, as shown in Figure 12. The residual structure consists mainly of a two-layer convolutional network and a nonlinear mapping process. Weight Norm [35] is a data normalization method that rewrites parameters and is often used to accelerate model convergence. By normalizing the weight, it can ensure that the gradient range is suppressed when the gradient is backpropagated, and thus, the gradient becomes self-stabilizing. In addition, the Dropout layer can ignore a certain number of neurons to reduce the overfitting phenomenon. TCL contains five layers of residual blocks.

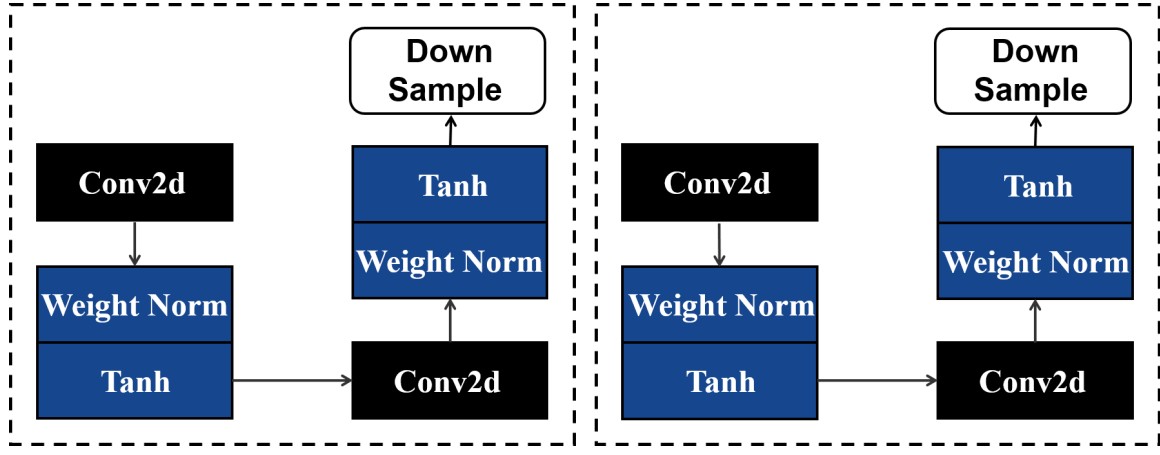

**Figure 12.** Multilayer residual structure.

## 4. Experiments

### 4.1. Experimental Setup

All experiments were performed and tested on the Windows 10 system (CPU: Intel(R) Xeon(R) W-2133 CPU @3.60GHz; GPU: NVIDIA GeForce RTX 2080 Ti). In this paper, the daily inputting data time was 12 time periods, that is, 1 h in total, and the time for the most recent data input was 12 time periods or 1 h. The traffic data were predicted over the next 12 time periods, or for the next hour. In TCL, the initial expansion factor was 2 and the convolution kernel size was 3. The experimental settings of the two TCL layers were the same. In GCN, the spatial feature extraction accuracy increased as the number of Chebyshev polynomial terms increased. However, a larger number of items will increase the training difficulty. In this article, Chebyshev's third-order polynomial was used, that is, K = 3. In the training phase, all experimental batch sizes were 32, and the learning rate was set to 0.001. All experiments used the Adam optimizer [36] to train the model, and the maximum number of training iterations was set to 1000.

### 4.2. Dataset

The dataset selected for this paper were traffic flow data on California freeways in the United States. The data come from the Performance Measurement System (PeMS, http://pems.dot.ca.gov/) (accessed on 30 June 2022) on California freeways in the U.S. and traffic data in PeMS is collected in real-time from over 39,000 detectors. These sensors cover the freeway systems of all major cities in California. PeMS is also an archived data user service (ADUS) containing more than a decade of historical analysis data which integrate a variety of information from Caltrans and other local agency systems. The dataset was accessed on 10 August 2020. In this paper, 97 detectors were selected from PeMS as nodes in the training set. For these 97 nodes, the traffic flow data from 1 April to 1 May 2020 (30 days in total) are selected as the training set with 8640 5 min time segments in total. To verify the model performance, the test set is used for the traffic flow from 1 May to 25 May 2020, a total of 25 days. The test set data consists of 7200 5 min time segments.

### 4.3. Evaluation Metrics and Baselines

This paper selected Mean Absolute Error (MAE) and Root Mean Squared Error (RMSE) as indicators to evaluate the model performance. The MAE value range was $[0, +\infty)$. When the predicted value was fully consistent with the actual value, the MAE calculation result was equal to 0, that is, a perfect model. The larger the MAE, the larger the error. The definition expression of MAE is Equation (18).

$$\text{MAE} = \frac{1}{n} \sum_{i=1}^{n} |\hat{y}_i - y_i| \tag{18}$$

RMSE is often used as a standard for measuring machine learning model prediction results and measures the deviation between the observed and true values. The RMSE value range is the same as that of MAE. Similarly, the larger the RMSE, the larger the error. The definition expression of RMSE is Equation (19).

$$\text{RMSE} = \sqrt{\frac{1}{n} \sum_{i=1}^{n} (\hat{y}_i - y_i)^2} \tag{19}$$

We selected the experimental results of the following four baselines to compare with the GSTNCNI experimental results:

- GCN [30]: Graph Convolutional Network, a special CNN model.
- T-GCN [16]: A Temporal Graph Convolutional Network for Traffic Prediction. T-GCN is used to capture temporal and spatial dependence, for both short-term and long-term prediction.

- STGCN [17]: Spatiotemporal Graph Convolutional Network, a deep learning framework for traffic prediction. STGCN applies a pure convolutional layer to simultaneously extract spatiotemporal information from the graph structure time series.
- ASTGCN [19]: Attention Based Spatial–temporal Graph Convolutional Network; a deep learning model for traffic flow prediction. ASTGCN uses a spatiotemporal attention mechanism to highlight the temporal and spatial features of input data; GCN is then used to extract the input data spatial features, and standard convolution is used to obtain the temporal features.

### 4.4. Comparison Experiments and Ablation Experiments

The GSTNCNI was placed under the same experimental conditions as the other four baseline models to predict the test set, and the MAE, RMSE, and prediction results were recorded. During the experiments, traffic flows were predicted for the next 12 time segments (i.e., 60 min). Table 1 displays the evaluation results of all traffic flow prediction models on the same dataset. It can be seen that the model outperforms other baselines in the same training environment. Compared with the baseline, GSTNCNI incorporates node correlation analysis to improve the model prediction accuracy.

**Table 1.** Results of the evaluation of different flow prediction models.

| Model | MAE | | | RMSE | | |
|---|---|---|---|---|---|---|
| | 5 Min | 30 Min | 60 Min | 5 Min | 30 Min | 60 Min |
| GCN | 20.61 | 24.19 | 26.45 | 25.74 | 28.10 | 29.68 |
| T-GCN | 15.74 | 18.25 | 21.08 | 20.36 | 24.63 | 27.97 |
| STGCN | 12.86 | 16.77 | 17.83 | 18.64 | 20.71 | 24.02 |
| ASTGCN | 10.97 | 13.56 | 15.14 | 17.33 | 19.68 | 22.41 |
| Ours | 8.38 | 11.23 | 12.67 | 14.56 | 16.89 | 20.50 |

### 4.4.1. Comparison of GSTNCNI with a Model Containing a Graph Convolutional Neural Network

In Figure 13, each time step on the x-axis is 5 min, and the y-axis represents the amount of traffic passing through within that time. This paper selects the traffic flow within 400 consecutive time steps as the prediction target to more intuitively show the time series changes in traffic flow. In the model prediction process, the traffic flow of the first 12 time periods was taken to predict the traffic flow of the subsequent 12 time periods, and the prediction results were then spliced in order of time to obtain 400 consecutive time step predictions. The traffic flow data were standardized by the Z-score normalization method, and the mean value after standardization was 0. The initial value of the parameter matrix of GCN in the compared models was randomly generated, so it can be considered that the predicted value starts from the mean value, then iterates and optimizes according to the loss function through the backpropagation gradient descent method, and gradually fits the real value. Therefore, the prediction curve is generally below the real curve. Figure 13 shows a comparison of the prediction results between GSTNCNI and four models containing graph convolutional neural networks on the same dataset. It can be seen that the GSTNCNI prediction results are better than those of the four baselines, the traffic flow data have strong spatial correlation, and GCN can fully extract the traffic flow data spatial features. However, traffic flow data, as typical time-series data, have a strong temporal correlation. Although GCN can fully extract the spatial traffic flow data features, it lacks any temporal correlation analysis. Therefore, the GCN prediction effect is the worst among the four baselines. T-GCN adds the temporal correlation analysis module to GCN and extracts the temporal traffic flow data features via the temporal correlation analysis module. By comparing the T-GCN and GCN prediction results in Figure 13, it can be seen that the prediction effect of the model after incorporating the time correlation analysis module is significantly improved. As shown in Table 1, the T-GCN MAE is reduced by 5.39 on average, and the RMSE is reduced by 3.52 on average compared with GCN. T-GCN uses CNN to extract traffic flow data temporal

features, but CNN is commonly used to extract Euclidean structured data spatial features, including images and speech signals. Therefore, CNN cannot fully analyze the traffic flow data temporal correlation. STGCN uses GRU to achieve traffic flow data temporal feature extraction. Compared with CNN, GRU, as a typical recurrent neural network model, has additional advantages in temporal correlation analysis. It can be seen in Figure 13 that the STGCN prediction effect is closer to the real value. It can also be seen from Table 1 that the STGCN MAE decreases to 15.82 on average and the RMSE decreases to 19.81 on average. ASTGCN incorporates the STGCN-based attention mechanism. The attention mechanism can effectively improve the efficiency of the spatial–temporal module in extracting the traffic flow data spatial–temporal features. However, the GCNs in all four baseline models use the traffic network adjacency matrix to calculate the Laplace matrix and complete the spatial correlation analysis. This method lacks traffic node feature analysis, so to address this problem, this paper uses the topological features of six complex networks to assist in calculating the Laplace matrix and improve the GCN. The traffic prediction comparison graph in Figure 13 shows that GSTNCNI can accurately fit the real traffic flow data and improve traffic flow data prediction. The numerical comparison plot in Figure 13 shows that the GSTNCNI prediction effect is uniformly distributed around the y = x diagonal, indicating that the error between the predicted and real values of GSTNCNI is smaller. Compared with baselines, the prediction results in this paper are closer to the real data.

### 4.4.2. Ablation Experiments

Figure 14 shows a comparison of the GSTNCNI prediction results with the model that retains only one complex network characteristic. Among them, Ours_DC indicates a model that only incorporates degree centrality; Ours_CC indicates a model that only incorporates clustering coefficients; Ours_CTC indicates a model that only incorporates closeness centrality; Ours_BC indicates a model that only incorporates betweenness centrality; Ours_PS indicates a model that only incorporates point strength; Ours_ASPLN indicates a model that only incorporates the average shortest node path length; Ours indicates a model that incorporates all six of the above complex network features. Ours_None indicates a model that does not contain the above features. By analyzing Figure 14, it can be seen that the model incorporating only one network characteristic has errors in the traffic flow data prediction results. However, compared with the Our_ None, their prediction results all have improved. Among them, the model that only incorporates the average shortest node path length has the worst prediction results. The node average path length only contains part of the nodal distance information, which has less influence on the spatial correlation analysis. Furthermore, the model that only incorporates clustering coefficients has inferior results. The clustering coefficients reflect the degree of aggregation between nodes in the graph. The model incorporating the clustering coefficients can fully extract the Euclidean structure data spatial features. However, as traffic flow consists of typical non-Euclidean structure data, the clustering coefficients cannot fully analyze the traffic flow data spatial features. As a result, the Ours_CC prediction results still contain large errors compared with the real traffic flow data. The GCN model, which uses Chebyshev polynomials instead of a Fourier transform, achieves traffic flow data spatial correlation analysis with the help of the node degree matrix. Ours_DC calculates the node degree centrality based on the node degree matrix. The node degree centrality can effectively enhance the node correlation information and reduce the model prediction error. Although node degree centrality can effectively improve the traffic flow prediction accuracy of individual traffic nodes, it lacks global node correlation analysis. Therefore, only incorporating node degree centrality has a limited improvement in the model prediction capability. The node degree reflects the node network structure features, while the point strength adds the node correlation edge weight information to the degree value, so the node features reflected by the point strength are more accurate and comprehensive than the degree value. As can be seen from Figure 14, the prediction effect of the model incorporating only point strength is significantly improved compared with the first three complex network feature metrics.

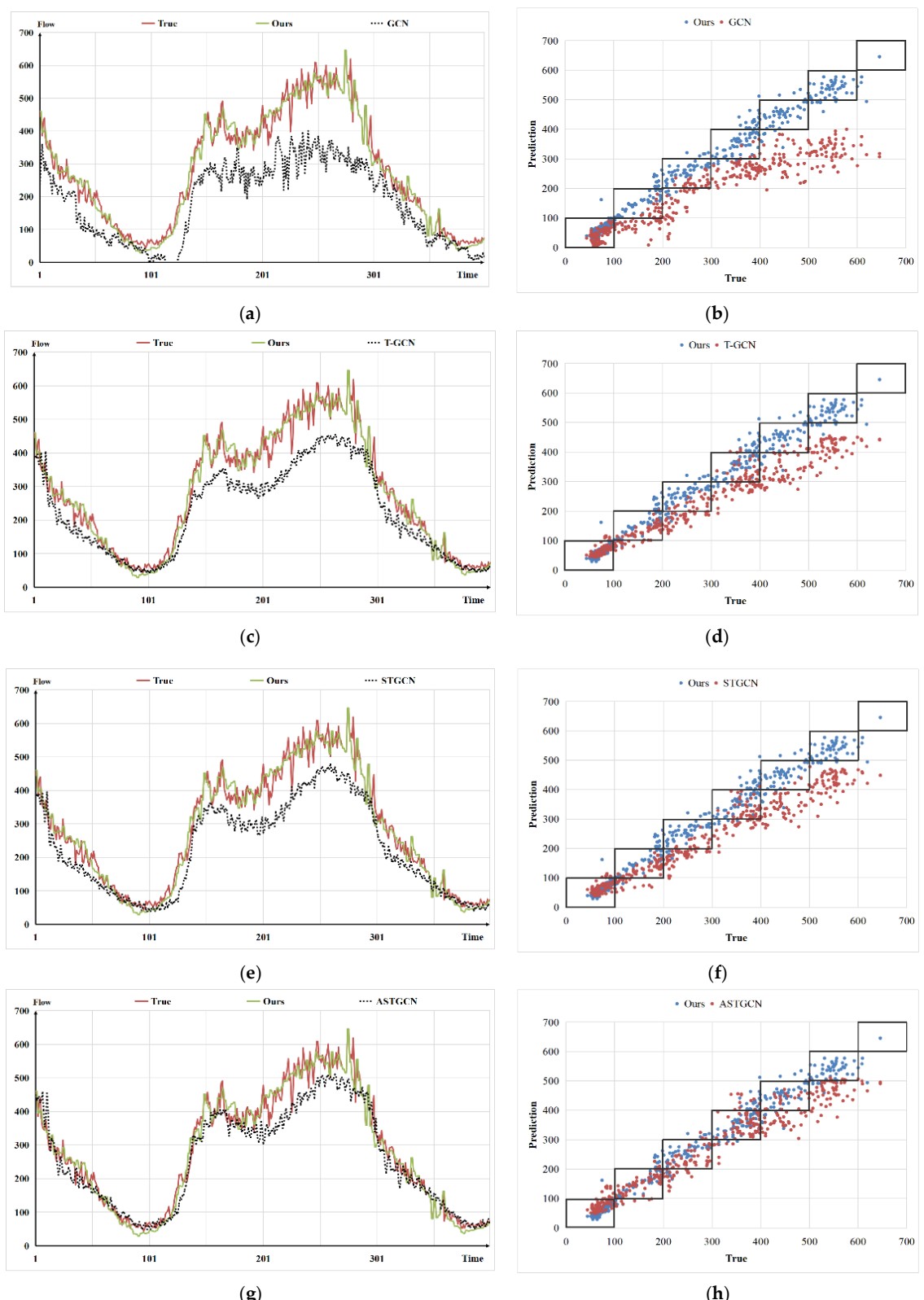

**Figure 13.** Comparison of the GSTNCNI experimental results with four models containing graph convolutional neural networks on the same data. (**a**) Comparison between GSTNCNI and GCN; (**b**) Scatter diagram of GSTNCNI and GCN prediction results; (**c**) Comparison between GSTNCNI and T-GCN; (**d**) Scatter diagram of GSTNCNI and T-GCN prediction results; (**e**) Comparison between GSTNCNI and STGCN; (**f**) Scatter diagram of GSTNCNI and STGCN prediction results; (**g**) Comparison between GSTNCNI and ASTGCN; (**h**) Scatter diagram of GSTNCNI and ASTGCN prediction results.

Although point strength can improve the model prediction, it lacks a global traffic network analysis. The Ours_CTC model uses closeness centrality to analyze global node features, and closeness centrality can obtain the node centrality degree based on the network global topology. By incorporating closeness centrality into the model, its analysis of global node correlation can successfully be improved and its effectiveness enhanced. Closeness centrality can reflect the centrality of a node in a complex network but lacks information about interactions between nodes. Betweenness centrality is a global statistic where two disjoint nodes in a network need to pass through other nodes to connect. The higher the number of times a node acts as an "intermediary", the greater its betweenness centrality. The betweenness centrality can adequately represent the node connectivity and incorporating betweenness centrality in the model can effectively improve its prediction ability. It can be seen from Figure 14 that the model only incorporating betweenness centrality has the best prediction among the six ablation experiments. The model incorporating the six complex network features can fully extract the local and global spatial features in the network and substantially improve the model prediction effect. By analyzing Figure 14, it can be seen that the prediction result of the model incorporating the six complex network features accurately fits the real traffic flow data variation.

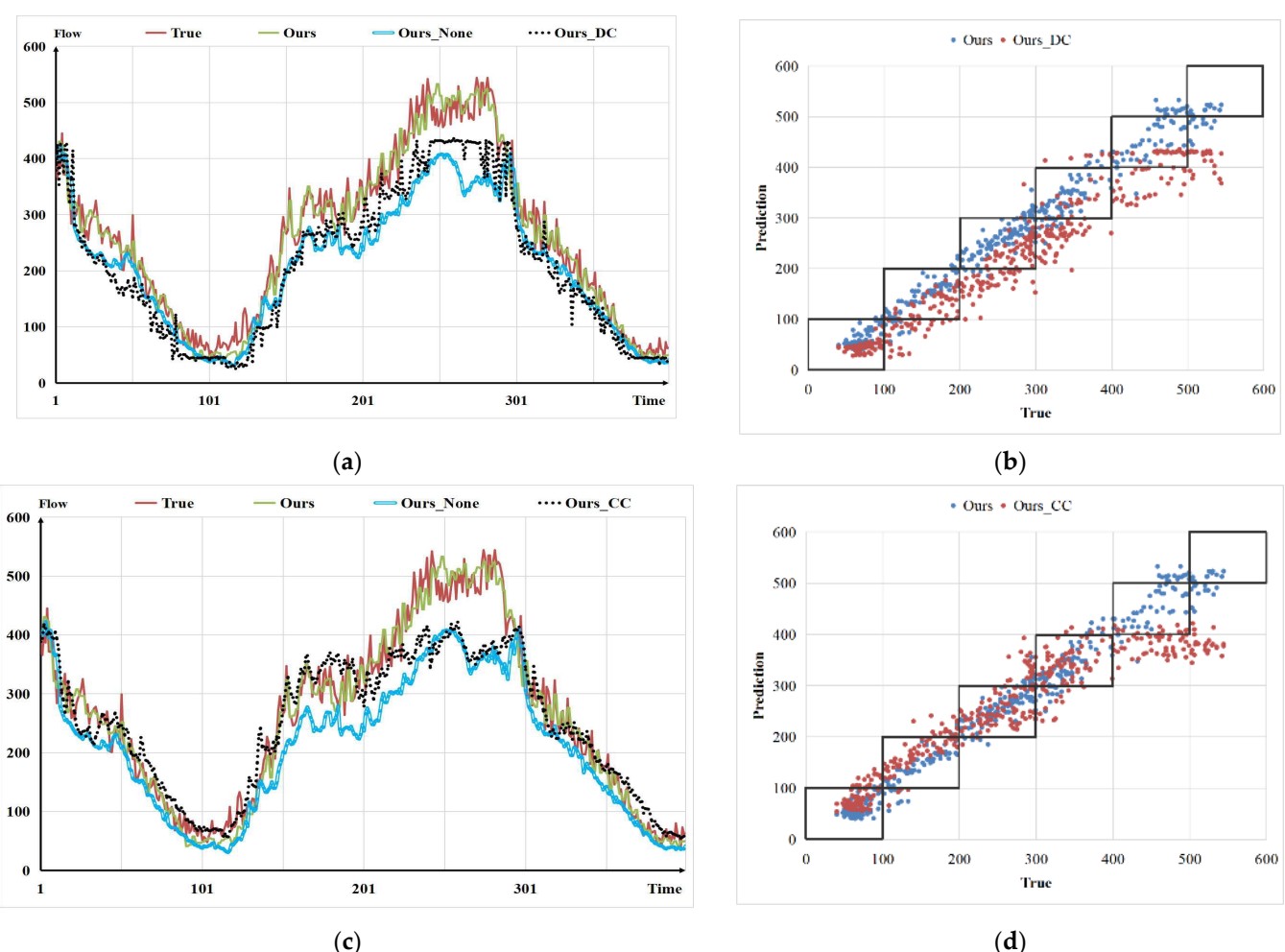

**Figure 14.** *Cont.*

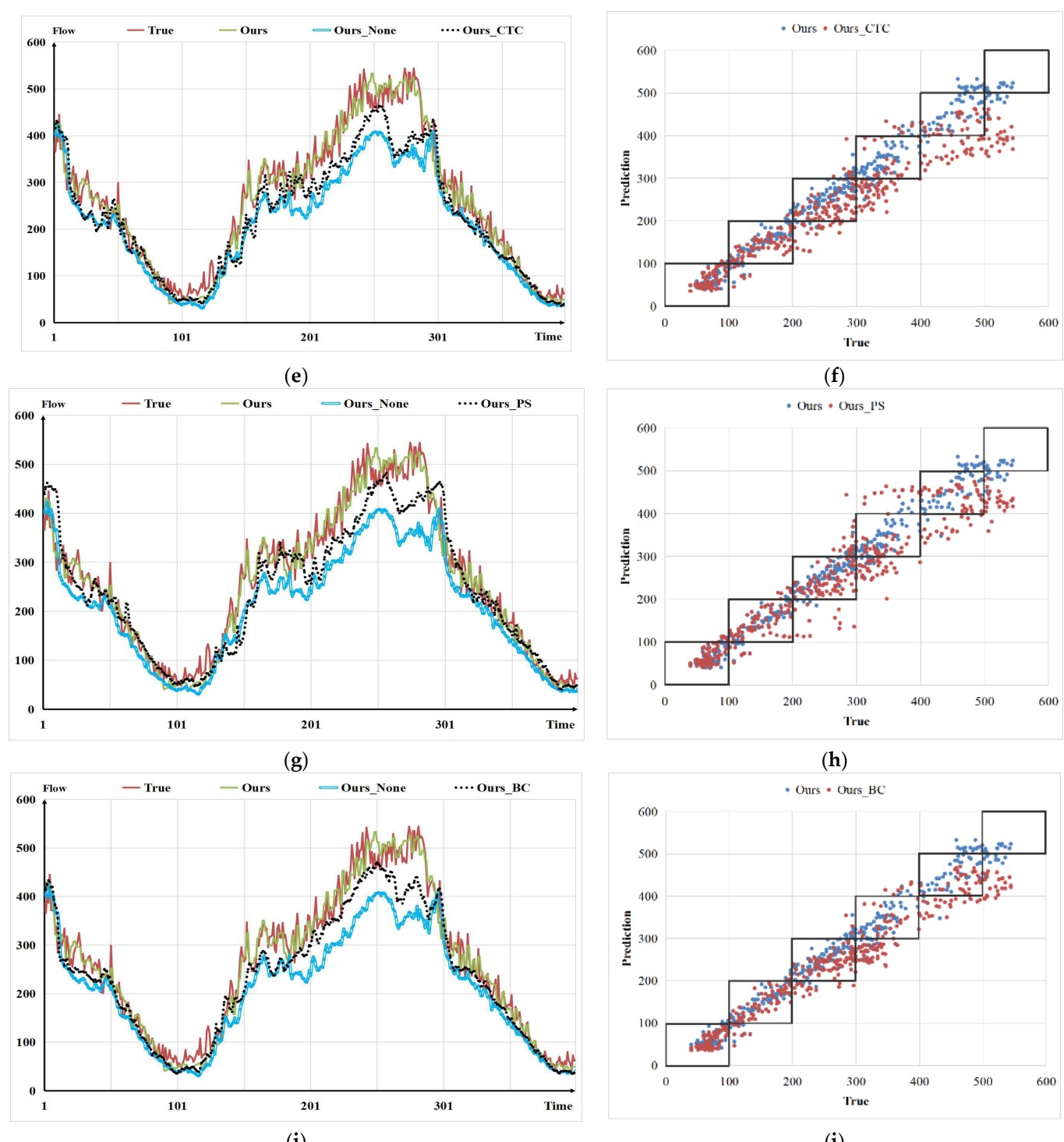

**Figure 14.** *Cont.*

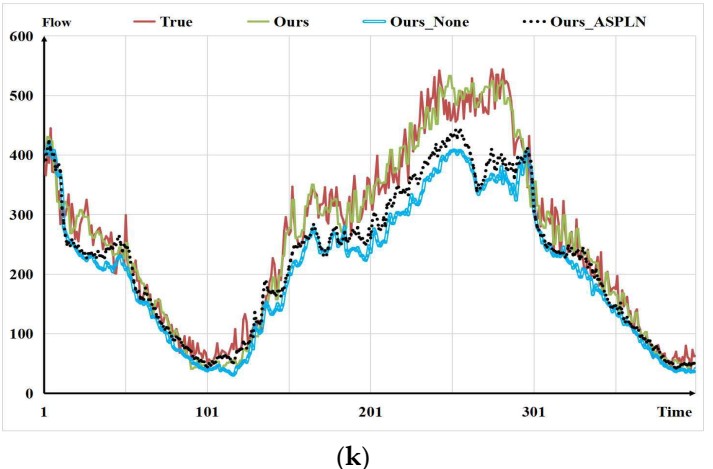

(**k**)

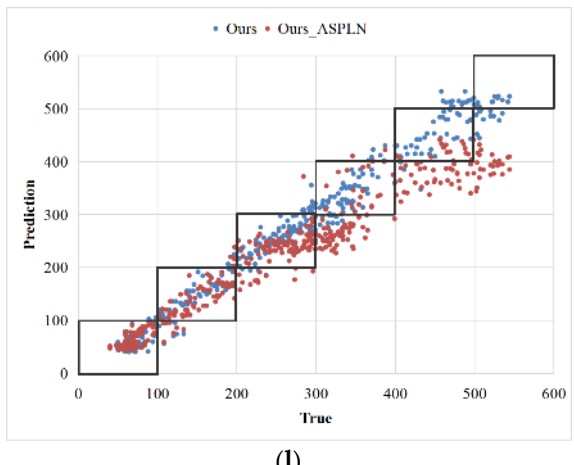

(**l**)

**Figure 14.** The model prediction results retain only one complex network characteristic compared with the GSTNCNI prediction results. (**a**) Comparison between GSTNCNI and Ours_DC; (**b**) Scatter diagram of GSTNCNI and Ours_DC prediction results; (**c**) Comparison between GSTNCNI and Ours_CC; (**d**) Scatter diagram of GSTNCNI and Ours_CC prediction results; (**e**) Comparison between GSTNCNI and Ours_CTC; (**f**) Scatter diagram of GSTNCNI and Ours_CTC prediction results; (**g**) Comparison between GSTNCNI and Ours_PS; (**h**) Scatter diagram of GSTNCNI and Ours_PS prediction results; (**i**) Comparison between GSTNCNI and Ours_BC; (**j**) Scatter diagram of GSTNCNI and Ours_BC prediction results; (**k**) Comparison between GSTNCNI and Ours_ASPLN; (**l**) Scatter diagram of GSTNCNI and Ours_ASPLN prediction results.

## 5. Discussion

Traffic flow data have a strong spatial correlation and certain baselines can adequately extract their spatial characteristics. However, traffic flow data, as typical time series data, also possess a strong temporal correlation. Although certain models can adequately extract the traffic flow data spatial features, they lack analysis of the traffic flow data temporal correlation. In recent years, graph spatial–temporal networks have been widely used in the field of spatial–temporal traffic flow data prediction. The unique features of graphs can capture the structural relationships between data. Therefore, more insights can be attained using graph structures rather than using data analysis. However, solving the learning problem on graphs is often challenging. An effective solution is to learn graph representations in low-dimensional Euclidean space to preserve certain graph attributes with the vast majority of the information. Deep learning models, based on graph structure information, have recently emerged in the deep learning field and have exhibited superior performance for a variety of problems. However, GCN uses the traffic network adjacency matrix to compute the Laplace matrix and complete spatial correlation analysis based on it. This method lacks analysis of traffic node characteristics and so to resolve this, this paper uses the topological features of six complex networks to assist in calculating the Laplace matrix and improve the GCN. The experimental results show that GSTNCNI can accurately fit real traffic flow data and improve traffic flow data prediction. Compared with all baselines, the complex network-based approach proposed in this paper improves the actual performance by about 31.46% on average.

In addition, the ablation experiments results show that models only incorporating one network feature have traffic flow data prediction errors. Among them, the prediction performance of the model that only includes the average shortest node path length is the worst. The average node path length only contains part of the node distance information and has little influence on spatial correlation analysis. Furthermore, the model that only includes the clustering coefficients performed poorly. Since traffic flow is typical non-European structure data, the clustering coefficient cannot fully analyze the spatial characteristics of traffic flow data. Node degree centrality can effectively enhance node-related informa-

tion and reduce model prediction errors. However, it lacks any analysis of global node correlations. The node degree reflects the node network structure characteristics, and the point strength adds the weight information of the node-related edges to the degree value. Consequently, the node characteristics reflected by the point strength are more accurate and comprehensive than the degree value. The addition of closeness centrality to the model effectively improves the global node correlation analysis, which helps improve the model's effectiveness. Betweenness centrality can fully represent the node connectivity and the addition of betweenness centrality to the model effectively improves the model prediction effect. In summary, the prediction results of the model combining the six complex network features can accurately fit the changes in real traffic flow data.

Finally, there are numerous specific application scenarios for this work. For example, Australia's most populated city, Sydney, in New South Wales, plans to spend millions of dollars to strengthen the monitoring and management of the region's road networks. The program called for Cubic to provide an intelligent traffic congestion management program, an operational example of a predictive analytics application. Sydney will utilize a data-driven model and management platform to predict traffic flow, reduce congestion, and respond to emergency traffic incidents on time. By the end of 2020, when the plan is complete, it will be the first city in the world to manage its transport network based on predictive analytical models. Despite the high level of complexity, predictive models are easily scalable. In other words, while the initial time and resource investment to build the base model are substantial, once completed, the model can be applied to cities, incrementally improving the quality of urban life.

## 6. Conclusions and Future Work

Complex network theory, as a new tool for graph structure data analysis, can deeply analyze and mine the spatial relationships of monitoring stations. In this paper, we firstly established a traffic complex network model by using traffic big data, then analyzed the topological features of the traffic road network via complex network theory, and finally combined the topological features with a graph neural network to explore the role of topological features in traffic flow prediction. Six complex network properties are discussed, namely, degree centrality, clustering coefficient, betweenness centrality, closeness centrality, point strength, and average shortest path length. In this paper, we improve the graph convolutional neural network based on the above six complex network properties and propose a graph spatial–temporal network that combines several complex network properties. By comparing with existing graph convolutional neural network baselines, it is verified that GSTNCNI has high accuracy and robustness in traffic flow prediction. In addition, ablation experiments are conducted on six different complex network features to verify the impact of different complex network features on the model's prediction accuracy. The experimental analysis shows that a model incorporating multiple complex network features has a more accurate prediction than a model incorporating only one complex network characteristic. The six complex network features analyzed in this paper can be divided into two main parts: locally relevant complex network features and globally relevant complex network features. Incorporating multiple complex network features can fully analyze local node correlation and global network correlation, which significantly improves the model prediction accuracy.

Although GSTNCNI can make accurate traffic flow data predictions, there are still several areas that can be improved. Firstly, the traffic data variation is not only affected by spatial and temporal correlation, but also by external influences, such as weather and holidays. Secondly, most graph neural networks are constructed based on static graphs, lacking the dynamic analysis of complex networks. Finally, existing models are based on short-term traffic data prediction and lack long-term data prediction ability. Consequently, we will continue to improve the related work. On the one hand, we consider the influence of external influencing factors on the model prediction effect, and on the other, we improve the model's long-term data prediction accuracy. Thus, traffic prediction tasks can be completed

accurately and efficiently, providing a scientific basis for the rational planning of traffic routes while improving travel efficiency.

**Author Contributions:** Conceptualization, Z.H., R.S. and F.S.; Methodology, Z.H. and R.S.; Software, Z.H.; Validation, Z.H.; Formal analysis, Z.H.; investigation, Z.H. and F.S.; Resources, R.S. and F.S.; Data curation, Z.H.; Writing–original draft, Z.H.; Writing review & editing, Z.H. and R.S. Supervision, F.S.; Project administration, R.S.; All authors have read and agreed to the published version of the manuscript.

**Funding:** This research received no external funding.

**Data Availability Statement:** The data used in the experiment can be obtained from the following links: https://github.com/hzqhappy/PeMS-97 (accessed on 30 June 2022).

**Conflicts of Interest:** The authors declare no conflict of interest.

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
