# Peer review of "A New Perspective on Traffic Flow Prediction: A Graph Spatial-Temporal Network with Complex Network Information"

_electronics, doi:10.3390/electronics11152432_

Round 1

Reviewer 1 Report

The paper presents a new architecture for traffic prediction, GSTNCNI, that employs complex road network graph feature information. The idea is quite interesting and seems to be original. The experiments are well designed and the paper can be valuable to the scientific community. In general, the article is well-written, but there are some issues that should be fixed before publication:

1) lower-case / upper-case issues - surprisingly, there are many such issues, e.g.,: 
- p. 2: yang et al -> Yang et al

- p. 2: ma et al [12] -> Ma et al [12]

- p. 2: "features. traffic data" -> "features. Traffic data"

- p. 8: ". the GSTNCNI" -> ". The GSTNCNI"
- p. 9: ". the TCL" -> ". The TCL"
- p. 10: "rewriting. weight Norm" -> "rewriting. Weight Norm"
- p. 12: ". compared with" -> ". Compared with"
- p. 13: ". traffic flow" -> ". Traffic flow"
- p. 16: ". the Ours_CTC" -> ". The Ours_CTC"
2) Lack of references - I recommend adding more references, for example, to:
- "However, with the rapid socio-economic development, car ownership keeps increasing, which in turn brings tremendous pressure on road traffic and traffic congestion is becoming more and more prominent." - it would be good to support it with a reference

- the PeMS dataset
- Space L and Space P
- Weight Norm
- Adam
- p. 11: GCN: Graph Convolutional Network, a special CNN model.
- p. 11: T-GCN
- p. 11: STGCN
- p. 11: ASTGCN
3) Wrong references to figures:
- p. 13-14: Figure 17 - there is no such figure
- p. 16: Figure 18 - there is no such figure
4) Issues with indices: 
- p. 3: "a is shown in Equation (1)" -> "a_{ij} is shown in Equation (1)"
- p. 5: "a node is Ki" -> "a node is K_i"
- p. 5: "between Ki nodes" -> "between K_i nodes"
- p. 5: "of edges between Ei nodes" -> "of edges between E_i nodes"
5) Issue with naming:
- p. 5: "aggregation coefficient" - it is in the section about "clustering coefficient" - is it the same? If not, "aggregation coefficient" is not defined.
- besides: how to interpret the aggregation coefficient equal to 0? According to Equation (4), it would mean that there are no edges (E_i = 0). Is it possible for 50% of sites? Looks strange.
6) Issues with a definition:
-  it is not sure what is the meaning of the index i in C_i, E_i. I believe this section should be revisited and prepared better.
- we have g_st ad g_st(i) in Equation (6) but they are not defined earlier (there is sth like "number of shortest paths through the node to the number of all shortest paths in the network." in the text, but it would be good to add g_st and g_st(i) in the text to be sure).
- p. 8: "degree matrix" - I believe it should be explained. Also, it appears earlier in Equation (9), so it should be explained earlier and Equation (9) should be explained in a clearer way (e.g., give a reference to info about Laplace matrix - some people may know it as Laplacian matrix)
7) Explanation of methodology:
- there is a sentence: "Where x denotes the input sequence; y denotes the output sequence," - this is a place where it would be good to explicitly explain what's the input and what's the output. Also, x_t was not formally defined, but is used later in (13).
- there is a description of the architecture, but it might be good to add some info about the number of layers of each type in the final architecture
- I recommend explaining better the part about Spatial Convolution Layer - it's crucial to understand the whole methodology - I've already suggested giving a reference (at least) to more info about the Laplace matrix and defining the degree matrix - maybe some additional figures would also help explain this part better.

8)  Analysis of results:
- it's quite interesting that almost all the compared models mostly underestimate the traffic flows - I believe it would be good to explain it, because maybe there is a systematic error done in experiments - maybe there are some reasons behind it, but I would welcome a discussion on this topic
- ablation study: instead of incorporating only 1 feature, it would be probably even more interesting to investigate what will happen if only one of the features is not incorporated - this may help to understand better if all the features are really needed

9) Typos / other issues:
- p. 8: "graph neural network is essential" -> "graph neural network it is essential"
- p. 7: "that lie" -> "that lies"
- p. 10: "Where, d denotes" -> "Where d denotes"

Reviewer 2 Report

The manuscript entitled “A New Perspective on Traffic Flow Prediction: A Graph Spatial-Temporal Network with Complex Network Information” has been investigated in detail. The topic addressed in the manuscript is potentially interesting and the manuscript contains some practical meanings, however, there are some issues which should be addressed by the authors:

1)      In the first place, I would encourage the authors to extend the abstract more with the key results. As it is, the abstract is a little thin and does not quite convey the interesting results that follow in the main paper. The "Abstract" section can be made much more impressive by highlighting your contributions. The contribution of the study should be explained simply and clearly.

2)      The readability and presentation of the study should be further improved. The paper suffers from language problems.

3)      The “Introduction” section needs a major revision in terms of providing more accurate and informative literature review and the pros and cons of the available approaches and how the proposed method is different comparatively. Also, the motivation and contribution should be stated more clearly.

4)      The importance of the design carried out in this manuscript can be explained better than other important studies published in this field. I recommend the authors to review other recently developed works.

5)      What makes the proposed method suitable for this unique task? What new development to the proposed method have the authors added (compared to the existing approaches)? These points should be clarified.

6)      “Discussion” section should be added in a more highlighting, argumentative way. The author should analysis the reason why the tested results is achieved.

7)      The authors should clearly emphasize the contribution of the study. Please note that the up-to-date of references will contribute to the up-to-date of your manuscript. The study named- A new hybrid model for wind speed forecasting combining long short-term memory neural network, decomposition methods and grey wolf optimizer; Detection of solder paste defects with an optimization‐based deep learning model using image processing techniques- can be used to explain the method in the study or to indicate the contribution in the “Introduction” section.

8)      How to set the parameters of proposed method for better performance?

9)      The complexity of the proposed model and the model parameter uncertainty are not enough mentioned.

10)  It will be helpful to the readers if some discussions about insight of the main results are added as Remarks.

This study may be proposed for publication if it is addressed in the specified problems.

Reviewer 3 Report

This paper presents a traffic flow prediction method, which is evaluated using open-source traffic data. Although the work is interesting, the writing should be improved. For some information, the reader should not wait until the end of the paper. Followings my comments to consider in the revision.

What kind of information is predicted from traffic? From what kinds of network, highways or urban roads with traffic lights? What is the predicted item, vehicles per hour or density, or travel time? From how many points of the network? The Abstract and Introduction do not contain any such information. Without explaining the target traffic prediction problem briefly, it is not easy to understand the contribution.

The four main contributions claimed in Introduction are not specified in the context of traffic. Why it is necessary to have a new traffic prediction model? What are the new features that make this better than the existing prediction mechanism? Is the prediction for 1 min, 5 min, or 10 min time? Or other intervals? They should be stated with contributions.

The target network California highway network should be shown in a graph with the marks of the nodes. In Figure 1, the network is plotted in 2D space. Are they corresponding to the physical distance of the sensors? Why it is not plotted on the California highway network? How the connectivity among nodes is obtained in the shown network?

The meanings of symbols used in the equation should be explained at their first use. Node degree values can be understood better with a physical network graph. Since the degree can be calculated directly, why it is necessary to show them in a probability graph? The same for betweenness, point strength and others. Does the connectivity or sensor position change or have uncertainty? If they are deterministic, the probability terms are confusing. How such probabilities are obtained?

In Figure 9, state the details of the inputs and outputs of the model.

What are the units of time and flow in the graphs? Why is the time points more than 288? Figure 12: what is the prediction time? Are they all predicted 1 hour in advance?

Finally, what would be the possible application of this method? To what extent do the instantaneous results differ from the historical average?

Round 2

Reviewer 2 Report

All my comments have been thoroughly addressed. It is acceptable in the present form.

Reviewer 3 Report

The authors have revised the paper.  Since this paper studies traffic in some specific urban areas, the reviewer recommends including the area maps along with the node points indications (or at least some node points, if all cannot be shown). Such a study area map can also be placed in Appendix, if necessary.  
